# A Peptide Inhibitor of NADPH Oxidase (NOX2) Activation Markedly Decreases Mouse Lung Injury and Mortality Following Administration of Lipopolysaccharide (LPS)

**DOI:** 10.3390/ijms20102395

**Published:** 2019-05-15

**Authors:** Aron B. Fisher, Chandra Dodia, Shampa Chatterjee, Sheldon I. Feinstein

**Affiliations:** Institute for Environmental Medicine, Department of Physiology, University of Pennsylvania Perelman School of Medicine, Philadelphia, PA 19104, USA; cdodia@pennmedicine.upenn.edu (C.D.); shampac@pennmedicine.upenn.edu (S.C.); sif@pennmedicine.upenn.edu (S.I.F.)

**Keywords:** peroxiredoxin 6, acute lung injury, sepsis, reactive oxygen species (ROS), PLA_2_ inhibitory peptide 2 (PIP-2)

## Abstract

We have previously derived three related peptides, based on a nine-amino acid sequence in human or rat/mouse surfactant protein A, that inhibit the phospholipase A_2_ activity of peroxiredoxin 6 (Prdx6) and prevent the activation of lung NADPH oxidase (type 2). The present study evaluated the effect of these Prdx6-inhibitory peptides (PIP) in a mouse (C57Bl/6) model of acute lung injury following lipopolysaccharide (LPS) administration. All three peptides (PIP-1, 2 and 3) similarly inhibited the production of reactive O_2_ species (ROS) in isolated mouse lungs as detected by the oxidation of Amplex red. PIP-2 inhibited both the increased phospholipase A_2_ activity of Prdx6 and lung reactive oxygen species (ROS) production following treatment of mice with intratracheal LPS (5 µg/g body wt.). Pre-treatment of mice with PIP-2 prevented LPS-mediated lung injury while treatment with PIP-2 at 12 or 16 h after LPS administration led to reversal of lung injury when evaluated 12 or 8 h later, respectively. With a higher dose of LPS (15 µg/g body wt.), mortality was 100% at 48 h in untreated mice but only 28% in mice that were treated at 12–24 h intervals, with PIP-2 beginning at 12 h after LPS administration. Treatment with PIP-2 also markedly decreased mortality after intraperitoneal LPS (15 µg/g body wt.), used as a model of sepsis. This study shows the dramatic effectiveness of a peptide inhibitor of Prdx6 against lung injury and mouse mortality in LPS models. We propose that the PIP nonapeptides may be a useful modality to prevent or to treat human ALI.

## 1. Introduction

Acute lung injury (ALI) is a syndrome manifested by diffuse damage to lung cells resulting in alteration of the alveolar permeability barrier with subsequent lung edema and altered gas exchange [1]. Although diverse etiologies can be responsible for the syndrome, there is general consensus that lung inflammation plays a major role in the pathogenesis of ALI for all etiologies, either as a primary factor in the initiation of lung injury or as an amplification mechanism [2,3,4]. Lung injury associated with inflammation is largely the result of agents released from inflammatory cells including reactive oxygen species (ROS) that, when produced in excess, can oxidize tissue macromolecules (lipids, proteins, DNA), resulting in widespread cellular destruction [3,5,6]. ROS can be produced in the lung through several mechanisms, including (a) auto-oxidation reactions, involving the mitochondrial respiratory chain or various other molecules such as quinones and ferrous iron; (b) as a by-product of the enzymatic activity of xanthine oxidase; and (c) through enzymatic activity of NADPH oxidase (NOX) enzymes [7,8,9]. The latter is a widely distributed family of seven enzymes that utilize NADPH to generate either superoxide anion (O_2_^•−^) or H_2_O_2_ as a primary product [10,11]. Unlike the auto-oxidation reactions, enzymatic reactions can be regulated and, thus, are likely to be significantly more important than auto-oxidation in physiological homeostasis. It is now known that ROS generated by NOX enzymes are crucial for the regulation of many important cell functions such as host defense, cellular signaling, cell migration, cell differentiation, and post-translational protein processing [11,12].

NOX2 was the original NOX family member to be described and has been the best studied from both biochemical and physiological standpoints [13,14]. NOX2 is a major source of ROS in polymorphonuclear leukocytes (PMN), alveolar macrophages (AM), and pulmonary endothelial cells (EC), and its presence at lower levels has been postulated in other lung cell types. In addition to its physiologic roles, excessive activation of NOX2 in lung cells has been associated with tissue injury and, in particular, may have an important pathophysiologic role in ALI as demonstrated with animal models using NOX2 “knock out” mice or various (albeit relatively non-specific) inhibitors of this enzyme [15,16,17,18,19,20,21,22]. NOX2 has been shown in mouse and rat models of bacterial lipopolysaccharide (LPS)-induced ALI to be responsible for increased ROS generation and ROS-mediated lung lipid peroxidation [9].

NOX2 is generally quiescent and its activation requires a stimulus that first leads to activation of specific cytoplasmic factors that, in turn, associate at the cell membrane with NOX2 and activate its ROS production. These cytoplasmic factors include the proteins p67^phox^, p47^phox^, and p40^phox^, as well as one of the small signaling G proteins, Rac 1 or 2, depending on the cell type. Commonly used activators of NOX2 for in vitro studies include angiotensin II (Ang 2) and phorbol esters such as phorbol myristic acetate (PMA) [23,24]. Specific physiologic stimuli for activation of this pathway during inflammation include bacterial products such as the peptide formyl Met-Leu-Phe, various cytokines, and a signaling cascade associated with LPS [7,25]. We have recently described the pathway from an initiating signal that activates the phospholipase A_2_ (PLA_2_) activity of Prdx6 leading to activation of the crucial Rac protein in PMN, AM, and EC [23,24,26,27]. This PLA_2_ activity is called aiPLA_2_, based on its activity at acidic pH and Ca^2+^ independence. Although Prdx6 also may be involved in the activation of NOX1 [28], the mechanism for this latter effect is unclear and it apparently does not involve the aiPLA_2_ activity of the protein. Thus, NOX activation via the aiPLA_2_ activity of Prdx6 appears to be specific for NOX2.

Our earlier studies related to regulation of Prdx6 activity indicated that surfactant protein A (SP-A) binds to the Prdx6 enzyme and inhibits its PLA_2_ activity [29,30]. Since both proteins (SP-A and Prdx6) are present in the lung lamellar bodies, the site of lung surfactant storage and metabolism, we postulated that regulation of the PLA_2_ activity of Prdx6 could be important for regulating the turnover of lung surfactant phospholipids. We next identified the respective binding sites in SP-A and Prdx6 that are responsible for their interaction, synthesized the respective peptides, and showed marked inhibition of Prdx6-PLA_2_ activity by the SP-A peptide without an effect on Prdx6 peroxidase activity [31]. More recently, we have evaluated the minimal effective sequence for maximal inhibition of aiPLA_2_ activity and identified three nonapeptides with similar inhibitory effect that we have called Prdx6-PLA_2_ inhibitory peptides (PIPs) [32]. These peptides are PIP-1 and -2—based on the mouse/rat and human SP-A sequences, respectively—and PIP-3, representing a hybrid between PIP-1 and -2. As the peptides do not cross cell membranes, they are encapsulated in liposomes for their intracellular delivery. All three peptides (in liposomes) effectively inhibit intracellular aiPLA_2_ activity and block NOX2 activation in lungs when administered by either the intratracheal (IT) or intravenous (IV) route. As all three peptides were effective as inhibitors of the Prdx6 target activity, we chose the human-based PIP-2 sequence for study of the LPS-treated mouse model in this manuscript because of its greater relevance for treatment of human disease.

Administration of LPS or endotoxin, a lipoglycan isolated from the cell membrane of gram negative bacteria, has been widely used as an animal model for ALI [33]. Intratracheal administration of LPS results in lung inflammation with alteration of alveolar capillary permeability and lung edema [27]. For the present study, we proposed that treatment of mice with PIP-2 would prevent NOX2 activation, inhibit ROS production, and ameliorate lung injury that is associated with intratracheal administration of LPS. We further investigated the effect of PIP-2 on mouse survival after the intraperitoneal injection of LPS, a model for sepsis-induced lung injury [34,35,36].

## 2. Results

### 2.1. Inhibition of Lung ROS Production by PIP

In order to confirm the inhibitory effect of PIP compounds on NOX2 activation, we studied ROS production by the isolated perfused lung in the presence of Ang II, a known activator of NOX2. Amplex red oxidation was used as an index of ROS production. There was a very low baseline rate of ROS production in perfused lungs under control conditions, that is, no added stimulant of NOX2 activity (Figure 1, wild type (WT) basal). ROS production was markedly increased with the addition of Ang II to the perfusate to activate NOX2 (Figure 1, WT control). ROS generation was decreased by 76% in NOX2 null compared to WT lungs, indicating that NOX2 is the major source of ROS entering the perfusate after Ang II stimulation. The addition of PIP-2 (in liposomes) to WT lungs inhibited ROS production (~75%) similar to NOX2 null, as shown previously [32]. Thus, PIP-2 resulted in essentially total inhibition of NOX2-mediated ROS generation. 

We next determined the effect of PIP-2 (in liposomes) on aiPLA_2_ activity and ROS production in lungs following LPS. These parameters were determined at 6, 12, and 24 h after the administration of IT LPS. aiPLA_2_ activity in the lung homogenate increased by ~50% compared to control at 6 h after treatment with LPS and increased by another 50% at 12 and 24 h (Figure 2A). We used an intracellular fluorophore (DFF-DA) to determine lung ROS generation. ROS-induced fluorescence was very low in the non-LPS-treated control lung but was increased ~10-fold at 6 h and ~20-fold at both 12 and 24 h in the LPS-treated mouse lungs (Figure 2B). This increase in lung DFF fluorescence after LPS could be slightly underestimated due to signal dilution from the presence of edema in these lungs (see below). Pre-treatment of mice with PIP-2 before LPS administration resulted in a dramatic decrease in aiPLA_2_ activity and in ROS-generated fluorescence at all three time periods to values similar to the non-LPS-treated control. These results indicate that intratracheal administration of LPS results in increased ROS production in the lung that is maintained for at least 24 h and can be inhibited almost totally by pre-treatment of lungs with PIP-2.

### 2.2. Time Course for LPS-Mediated Lung Injury

The sensitivity to LPS-mediated injury varies significantly among mouse strains [28]. For this study, we determined the course of lung injury for C57Bl/6J mice given IT LPS at 5 µg/g body weight (Figure 3). Lungs showed considerable injury when evaluated at 12 h after LPS as indicated by increased nucleated cells in the BALf, increased BALf protein, and increased wet to dry lung weight ratio (*p* < 0.05).These results are compatible with lung inflammation (cells in BALf), alteration of the alveolar-capillary permeability barrier (BALf protein), and lung fluid accumulation (lung wet/dry weight). The increase in lung tissue TBARS, 8-isoprostanes, and protein carbonyls indicates oxidative stress with oxidation of lung tissue lipid and protein components. These indices of lung injury showed similar values at 12, 16 or 24 h after LPS (Figure 3), indicating that the degree of lung injury was essentially stable at 12–24 h after this non-lethal dose of LPS. Partial recovery (~50%, *p* < 0.05) in the indices of lung injury was seen at 48 h although they were still elevated compared with control (*p* < 0.05).

### 2.3. Effect of PIP-2 on LPS-Mediated Lung Injury

To study the effect of PIP-2 administration on lung injury, mice were treated with LPS (5 µg/g body weight) given IT. The dose of LPS was chosen based on our previous studies using the same batch of LPS that showed a relatively low level of lung injury with 1 µg/g body weight and greater injury with no significant mortality using 5 µg LPS/g body weight [27,37]. PIP-2 (2 µg/g body weight in liposomes) was administered at 0, 12, or 16 h after LPS. We have previously shown that this dose of PIP-2 can inhibit lung aiPLA2 activity by ~90% for at least 24 h [32]. PIP-2 was given IT at time zero and IV at 12 or 16 h in order to avoid excessive damage to the trachea. Animals were sacrificed and lungs were examined at 24 h after LPS. All indices of lung injury, reflecting lung inflammation, alveolar-capillary barrier dysfunction, lung fluid accumulation, and tissue oxidative stress, were elevated in LPS-treated mice as compared to control (*p* < 0.05). PIP-2 administered at 0 time completely prevented lung injury when assessed at 24 h after LPS (Figure 4). Indices of tissue injury in lungs of mice treated with PIP-2 at 12 and 16 h also were markedly decreased compared to LPS alone and values were not significantly different from control values (Figure 4). Since lung injury was present in lungs at 12 and 16 h after LPS, the normal values at 24 h in lungs from LPS treated mice given PIP-2 at 12 or 16 h can only mean that lungs were able to recover fully from their injury during the 8 to 12 h interval between the administration of PIP-2 and examination of the lungs.

### 2.4. PIP-2 Treatment Prevents Mouse Mortality with High Dose LPS

Although mice treated with low dose LPS (5 µg/g body weight) suffer significant lung injury, it is transient and essentially all mice will recover from the insult (not shown). In order to test the effect of PIP-2 treatment with a more severe injury model, mice were administered a higher dose of LPS (15 µg/g body weight). At 12 h after LPS, mice were divided into 2 groups that received PIP-2 (in liposomes) or liposomes alone (placebo). Additional treatments, either PIP-2 or placebo, were administered to surviving mice at 12 h after the initial dose and then at 24 h intervals (36, 60 and 84 h); mice were sacrificed at 108 h (i.e.,120 h after LPS administration). Results for mouse survival are plotted with the time of randomization (12 h after LPS) as 0 time (Figure 5). At this higher dose of LPS, mice that were treated with placebo showed 73% mortality during the 12 h after the start of treatment and 100% mortality by 36 h. PIP-2 treated mice showed only 17% mortality (83% survival) at 36 h after the start of PIP-2 treatment and had no further mortality during the period of observation. In addition to the effect on mortality, a marked difference was observed in the behavior of mice that had received PIP-2 after LPS with a return of most mice to normal physical activity by 12 h after receiving PIP-2 (see Appendix A). Indices of lung injury in treated mice that were sacrificed at 120 h after LPS showed no abnormality indicating recovery from the acute effects of LPS (Table 1).

We next evaluated the effect of PIP-2 in mice given LPS (15 µg LPS/g body weight) by the intraperitoneal route as a model for ALI associated with systemic sepsis. We chose the dose of LPS based on our previous study that showed 60% mortality with 10 µg LPS/g body weight [34]; our goal was to produce 100% mortality in the placebo-treated mice, similar to that seen with the IT LPS study. At 12 h after LPS, mice were divided into treatment groups for liposomes only (placebo) or PIP-2 as described for IT LPS. Survival of placebo-treated mice (liposomes only) at 12 h after start of treatment (24 h after LPS) was less than 40% and 100% of mice were dead by 36 h (Figure 6). By contrast, treatment of mice with PIP-2 (2 µg/g body weight) increased survival at 36 h after start of therapy (48 h after LPS) to ~80% and ~40% of mice fully recovered. With a higher dose of PIP-2 (20 µg/g body weight), the long term survival rate was significantly greater at ~70%. Thus, PIP-2 markedly increased mouse survival in this model of ALI associated with systemic sepsis.

## 3. Discussion

ALI is a serious disease syndrome with a mortality rate of ~40%. Inflammation is an important factor that can amplify the lung injury associated with the primary insult. To date, there is no approved pharmacologic treatment for the inflammatory component of the syndrome. The mechanisms for lung injury during lung inflammation are complex, but excessive ROS production appears to play a major role [2,3,4,5,6,7]. We have shown previously that the aiPLA_2_ activity of Prdx6 is required for activation of ROS production by NOX2 [24,26] and have described several nonapeptides derived from lung surfactant protein A (SP-A) sequences that inhibit aiPLA_2_ activity and thereby inhibit the activation of NOX2 in lung cells [32]. These peptides are called PLA_2_-inhibitory peptides (PIP-1, PIP-2, and PIP-3). Although PIP-2 appeared to be slightly more active than the other 2 peptides, all 3 PIP compounds were effective as inhibitors, presumably reflecting in part the high degree of conservation of the Prdx6 amino acid sequence among species [32]. The mechanism for aiPLA_2_ inhibition appears to be the binding of the SP-A–derived peptide to Prdx6 and we have demonstrated that the site for binding of the 16 amino acid precursor of the PIPs is to the sequence comprising amino acids 195 to 204 of Prdx6 [31]. The sequence for this segment of human Prdx6 is: 195-EEEAKKLFPK-204; the corresponding mouse sequence is the same for eight of the 10 amino acids with Q rather than K at position 200 and C rather than L at position 201 [32,38]. We chose PIP-2, the PIP that was derived from the human SP-A sequence, for subsequent investigations. The amino acid sequence of PIP-2 is LHDFRHQIL [32]. The present study confirms that PIP-2 inhibits ROS production by AngII-activated NOX2 in the isolated mouse lung.

The primary goal of the present study was to evaluate the effect of PIP-2 on lung injury associated with the intratracheal administration of LPS. We first demonstrated that PIP-2 markedly inhibited AngII-mediated ROS generation; AngII is a known activator of NOX2 and we have shown previously that AngII-mediated NOX2 activation requires aiPLA_2_ activity [24]. We then showed that treatment with LPS resulted in both a marked increase in aiPLA_2_ activity of the lungs and also a marked increase in ROS production through the activation of NOX2; both the LPS-mediated increase in aiPLA_2_ activity and increase in ROS production were inhibited by PIP-2.

The first experiments to evaluate PIP-2 in the lung injury model utilized the concurrent administration of PIP-2 with LPS which markedly protected against subsequent lung injury. Measurements to evaluate acute lung injury following LPS administration included (a) nucleated cells in BALf (inflammation); (b) protein in BALf (alveolar-capillary permeability); (c) lung wet to dry weight ratio (lung edema); and (d) lung TBARS, 8-isoprostanes, and protein carbonyls (oxidation of tissue lipids and proteins). All of these indices of injury were significantly elevated in lungs that were evaluated at 12–24 h after administration of LPS. However, none of these indices of tissue injury were altered in lungs when PIP-2 was administered concurrently with LPS. Thus, PIP-2 clearly can prevent ALI associated with LPS administration in mice.

The next experiments investigated the effect of PIP-2 administered at 12 or 16 h after administration of LPS as a treatment (as opposed to preventative) modality. As shown in Figure 3, the tissue injury associated with non-lethal LPS is maximal at these time points. The delayed treatment scenario is most relevant to human disease since the lung injury is what usually brings the patient to medical attention. With PIP-2 administration at either 12 or 16 h after LPS, parameters of lung injury had returned to essentially normal values when examined at 24 h after LPS (Figure 4). Our conclusion from this study is that PIP-2 prevented ongoing lung injury associated with LPS and allowed the lung to repair itself during the 8–12 h between PIP-2 administration and sacrifice of the animal.

Our final study was to evaluate the effects of PIP-2 on mouse lung function and survival following the administration of a lethal dose of LPS. PIP-2 administered every 12–24 h following administration of LPS led to a dramatic improvement of mouse behavior (Appendix A), markedly reduced mouse mortality (Figure 5), and resulted in return of the indices of lung injury to normal values (Table 1). Thus, the nonapeptide inhibitor of the PLA_2_ activity of Prdx6 prevented ROS generation subsequent to NOX2 activation and prevented mortality associated with the administration of a lethal dose of LPS. These results indicate that, in addition to preventing injury, PIP-2 can treat LPS-induced ALI in the mouse.

The present results with PIP-2 give a conclusion similar to that of our previous studies using various means to inhibit aiPLA2 activity and the subsequent activation of NOX2. The previously published methods were (a) administration of MJ33, a lipid compound that is a competitive inhibitor of aiPLA_2_ activity; (b) use of Prdx6 null mice (a less than perfect model since the peroxidase activity of Prdx6 also is lost); and (c) use of mice with mutation of amino acid D140 in Prdx6, an essential component of the aiPLA_2_ active site [27,34,37]. The MJ33 inhibited mouse, the D140A mutant mouse, and the PIP-2 treated mouse all retain the peroxidase activity of Prdx6 while this activity is abolished in the Prdx6 null mouse. In these previous studies, LPS was administered by the IT route in (a) and (b) as a model for direct lung injury and by the intraperitoneal route in c) as a model for non-infectious sepsis. We have proposed that the mechanism for the protection afforded by PIP-2 is its inhibition of the aiPLA_2_ activity of Prdx6 by allosteric effects resulting from binding of the peptide to Prdx6 [31]. The PIP peptides do not inhibit other lung PLA_2_ enzymes as demonstrated experimentally and as expected based on dissimilarity of potential binding sites on the different proteins [32]. The inhibition of aiPLA_2_ activity prevents the generation of lysoPC and its downstream products, thereby preventing the activation of Rac, a necessary co-factor for Nox2 activation [24]. Interestingly, the cholesterol-lowering drug simvastatin also inhibits the activation of Rac [39], and has been shown to inhibit ROS production by endothelial cells and to be protective in mouse models of LPS-induced ALI [40,41]. Although there is no definitive evidence as yet, it is possible that inhibition of Rac activation has salutary effects on non-ROS mediated manifestations of ALI in addition to its effect on NOX2 activation.

The present and previous studies have shown that NOX2 is a major source of ROS in lungs and that the enzyme is activated in the presence of LPS. In addition to the LPS model, ROS generation by NOX2 has been shown to play a central role in several other related as well as disparate animal models of ALI including gram negative sepsis [21], endotoxin [42], severe trauma [17], hemorrhagic shock [43], and oleic acid instillation [40]. Presumably, a major manifestation of the oxidant stress associated with NOX2 activation is the oxidation of tissue macromolecules as shown in the present study. However, another important pathophysiological role associated with NOX2-derived ROS is based on evidence that ROS are responsible for the signals leading to neutrophil recruitment to the lung and the resultant lung inflammation that is characteristic of ALI [41]. The marked decrease in nucleated cells in BALf after treatment with PIP-2 suggests that this function of ROS is important for the recovery from lung injury. In that respect, a peptide inhibitor of the myristoylated alanine-rich C kinase substrate (Marcks) protein also protects against lung injury with LPS in mice [44]. Although this latter peptide has not been shown to inhibit NOX2 activation, its effects may be mediated through altered cellular motility that prevents PMN influx into the lung. Thus, PIP-2, simvastatin, the Marcks protein inhibitor, and possibly inhibitors of NOX2 such as apocynin all may prevent PMN influx into the lung after LPS, thereby reversing inflammation and the associated lung injury.

Based on the present results, the peptide inhibitors of NOX2 activation could be effective as preventative agents for patients at risk for ALI as well as for treatment of patients with established ALI. Major advantages of PIP-2 compared to other proposed treatments include its presumed low toxicity as an endogenous amino acid sequence, its effectiveness with intravenous injection, its relatively long intracellular ½ life, and its effectiveness at a relatively low dose. Although toxicity of these small peptides is not expected based on their normal expression in lungs as a component of the SP-A protein, this must still be investigated. The antigenic potential of the peptide theoretically is low, but that will need to be confirmed in humans. Other possible side effects of the PIP peptides include those associated with inhibition of Rac activation as well as loss of the signaling and regulatory functions of ROS. Of note, no major effects have been reported as yet that could be related to the inhibition of Rac with the widely used drug, simvistatin. A potentially more important side-effect of treatment with PIP could be the effect of inhibited ROS production on the bactericidal activity of inflammatory cells (PMN and macrophages) that use superoxide anion generated through activity of NOX2 for the killing of bacteria [45]. Further, it has been shown that some antibiotics require ROS for maximal efficacy [46]. Despite the theoretical possibility of an altered response to infection, an inhibitor of NOX2 activation did not decrease bactericidal activity of PMN in an LPS model of ALI [41]. This may reflect the ability of non-NOX2 pathways to compensate for the loss of NOX2-derived ROS. Although this would emphasize the important role for antibiotic coverage in patients being treated with NOX2 inhibitors, it is important to note that the use of antibiotics alone has not been effective in reducing mortality with this disease to a value significantly below 40% [47].

## 4. Materials and Methods

### 4.1. Animals

C57Bl/6J or NADPH oxidase (Nox2) null mice were obtained from the Jackson Laboratories (Bar Harbor, ME, USA) and were maintained under HEPA-filtered air with 12 h light/dark cycles in the facilities of the University of Pennsylvania Laboratory Animal Resources (ULAR). All procedures involving mice have been continuously approved since 29 March 2012 (latest review 27 July 2018) by the University of Pennsylvania Institutional Animal Care and Use Committee (IACUC) and are in accordance with the Guide for the Care and Use of Laboratory Animals published by the US National Institutes of Health.

### 4.2. Reagents

Peptides were synthesized by Apeptide, Shanghai, China; the peptide sequences have been published previously [32]. The estimated purity of the peptides, evaluated by mass spectroscopy, was >89%. Lipopolysaccharide (LPS), derived from *Escherichia coli* 0111:B4 cell membranes and purified by gel-filtration chromatography, was obtained from Sigma-Aldrich (St. Louis, MO, USA, cat # L3012). The Amplex red/ horseradish peroxidase (HRP) assay kit (cat.#A22188) and the carboxy adducts of reduced difluorofluorescein diacetate (DFF-DA, cat.#13293) were purchased from Life Technologies, Grand Island, NY, USA (through Thermo-Fisher Scientific, Waltham, MA, USA). Angiotensin II (Ang II) was obtained from Bachem, Torrance, CA, USA (cat. #4095850.0005). Authentic lipids were purchased from Sigma-Aldrich, St. Louis, MO, USA and liposomes were prepared by evaporation to dryness followed by reconstitution in saline as previously described to reflect the composition of lung surfactant [27]; the liposome composition was, in mol fraction, 0.5 dipalmitoylphosphatidylcholine (DPPC), 0.25 egg phosphatidylcholine (PC), 0.10 phosphatidylglycerol (PG) and 0.15 cholesterol. PIP-2 when added was 0.15 µg PIP-2/µg lipid.

### 4.3. Administration of LPS and PIP-2

Anesthetized mice were administered LPS (either 5 or 15 µg/g body weight) in 20 µl saline that was instilled into the lung through an endotracheal catheter placed at the level of the tracheal carina. We have shown previously that PIP-2 is ineffective if injected alone, while it inhibits aiPLA2 activity with a ½ time of ~50 h if encapsulated in liposomes [32]. PIP-2 in liposomes was suspended in 20 µL saline for IV or IT injection. For studies to evaluate the effect of PIP-2 at zero time, LPS administration was followed by liposomes ± PIP-2 also given by IT instillation. For studies to evaluate the effect of PIP-2 administered at later times after LPS, the liposomes ± PIP-2 were given by injection into a retinal artery. This shift in route of administration was used to minimize damage to the mouse trachea that could occur with repeated tracheostomy and lead to untoward effects on the lung. The dose of PIP-2 used for treatment after intratracheal LPS was 2 µg/g mouse body weight; in control mice, this dose of PIP-2 has been shown to inhibit lung aiPLA_2_ activity maximally for at least 24 h [32]. We gave the second dose of PIP-2 at 12 h after the first dose to be certain of maximal coverage and then went to every 24 h for PIP-2 administration for the subsequent 3 doses (5 doses in all). For the model of sepsis, LPS (15 µg/g body weight) in 20 µL saline was injected intraperitoneally and mice were treated with IV PIP-2 at either 2 or 20 µg/g body weight beginning at 12 h after LPS. We used the same times of PIP-2 administration in the sepsis model as were used for the IT LPS model. After recovery from anesthesia, all mice were maintained in the vivarium with ad libitum access to food and water. These methods have been described previously [27].

### 4.4. Evaluation of Lung Injury

At the end of each experiment with IT LPS (at either 24 or 120 h), surviving mice were sacrificed by exsanguination under anesthesia. Lungs in situ were cleared of blood by perfusion through the pulmonary artery and then were lavaged through the trachea with saline. The lung was then removed from the thorax for tissue assays. We evaluated the effect of LPS on lung injury by measuring the number of nucleated cells and the protein content in the lung bronchoalveolar lavage fluid (BALf), the lung wet to dry lung weight ratio using the left upper lobe of lung, and thiobarbituric-acid reactive products (TBARS), 8-isoprostanes, and protein carbonyls in the lung homogenate to determine the oxidation of lung tissue lipid and protein components. The methods used for these assays all have been described previously [27,34,48]. For studies of mouse mortality, survival plots were constructed using the Kaplan-Meier estimator [49].

### 4.5. Measurement of Lung ROS Production and aiPLA_2_ Activity

The effect of the PIPs on ROS production in control (untreated) lungs was determined in vitro with isolated perfused lungs. PIP-2 in liposomes was administered at 2 µg/g mouse weight by the IV route. After 30 min, mice were anesthetized and lungs were isolated, cleared of blood, and perfused in a recirculating system with perfusate containing Ang II (50 µM) as a Nox2 activator and Amplex red plus horseradish peroxidase to detect ROS [23,24,26]. Lungs from wild type mice and lungs from NOX2 null mice that were not treated with PIP-2 were used as controls. The basal rate of ROS production was evaluated with WT lungs that were perfused in the absence of AngII. The perfusion protocol included a 15 min equilibration period followed by a 60 min experimental period. Aliquots of perfusate were taken at 15 min intervals and analyzed by fluorescence for resorufin (*λ*_excitation_ 568 nm, *λ*_emission_ 581 nm), the product of Amplex red oxidation [26,50]. The rate of Amplex red oxidation was calculated and expressed as arbitrary fluorescence units (AFU) with normalization to mouse body weight. There was a low rate of Amplex red oxidation in the absence of HRP in the perfusate (~7% of the AngII-stimulated fluorescence), indicating a non-ROS-mediated oxidation of the fluorophore; this value was subtracted to obtain the reported values.

To determine lung ROS production after LPS treatment, intact mice were treated with LPS (5 µg/g) ± PIP-2 (2 µg/g). Mice were anesthetized at 6, 12, or 24 h after treatment with LPS and lungs in situ were cleared of blood and then perfused for 10 mins with saline solution containing the fluorophore DFF-DA that is hydrolyzed intracellularly to DFF [51]. Lungs were then homogenized, and fluorescence of the homogenate was measured at *E*x 495 nm, *E*m 525 nm. Lung fluorescence was expressed as AFU per minute of perfusion with normalization to the mouse body wt.

### 4.6. Statistical Analysis

Data are expressed as means ± standard error (SE). The slope of linear plots was calculated by the least mean squares method. SigmaStat software (Jandel Scientific, San Jose, CA, USA) was used to assess statistical significance. Mean values for group differences were evaluated by 1-way ANOVA followed by the Bonferroni post hoc test. For comparison of two groups, means were compared by Student’s *t*-test. Differences between mean values were considered statistically significant at *p* < 0.05.

## 5. Summary and Conclusions

A nine amino acid peptide (PIP-2) derived from lung surfactant protein A (SP-A) prevented NOX2 activation and inhibited ROS production in mouse lungs. PIP-2, given at 12–16 h after administration of a non-lethal dose of LPS, allowed essentially complete repair of lung injury evaluated at 24 h after LPS. PIP-2 given to mice by repeated doses every 12–24 h after a lethal dose of LPS markedly protected against death and allowed full recovery from lung injury evaluated at five days after LPS. PIP-2 also increased mouse survival after a lethal dose of LPS administered intraperitoneally as a model of sepsis. Thus, PIP-2, or an analogue, may be a valuable adjunct for the prevention or treatment of ALI, certainly in mice and hopefully in humans. In that respect, it is important to note that ALI, by its nature, represents a multifactorial syndrome that clearly requires polytherapy for its control. Examples of polytherapy in current use include ventilatory support variously combined with antibacterial or antiviral therapy for associated infection, corticosteroids for a hypersensitivity component, vasopressors and/or blood transfusions for hypovolemia, etc. Since ALI due to any etiologic factor is likely to be associated with lung inflammation, the administration of PIP-2 could be an important adjunct to prevent the oxidative stress component of lung inflammation that is mediated by ROS release associated with activation of NOX2.

## Figures and Tables

**Figure 1 ijms-20-02395-f001:**
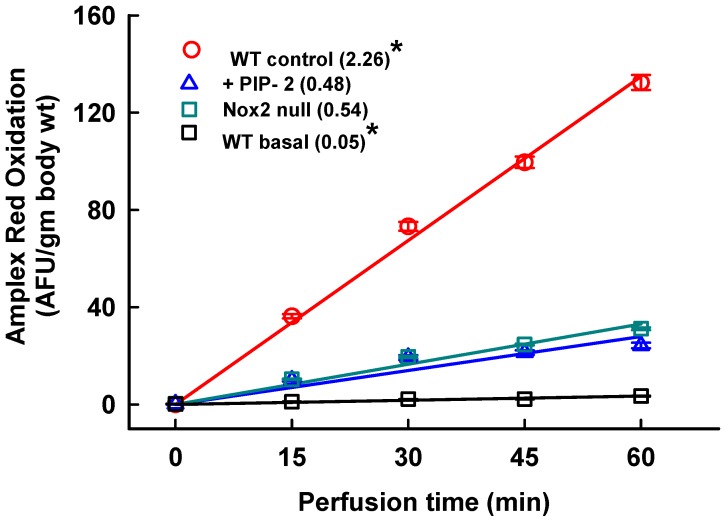
PLA_2_ inhibitory peptide (PIP-2) inhibits ROS production stimulated by angiotensin II (Ang II) in isolated perfused mouse lung. PIP-2 (2 µg/g body weight) was administered to intact wild type (WT) mice by the IV route. WT basal, WT control and NOX2 null did not receive peptide. After 30 minutes, lungs were isolated from anesthetized mice and perfused in a recirculating system with added Ang II (50 µM) as a Nox2 activator and Amplex red plus horseradish peroxidase to detect perfusate ROS. WT basal lungs were not stimulated with Ang II. After a 15 min equilibration period (called zero time), aliquots were taken at 15 min intervals for analysis of fluorescence. Each plotted point represents the mean ± SE for *n* = 3. The lines were drawn by the least mean squares method. The mean rates of ROS reduction calculated from the slope of each line are indicated in parentheses. * *p* < 0.05 vs. the other three slopes.

**Figure 2 ijms-20-02395-f002:**
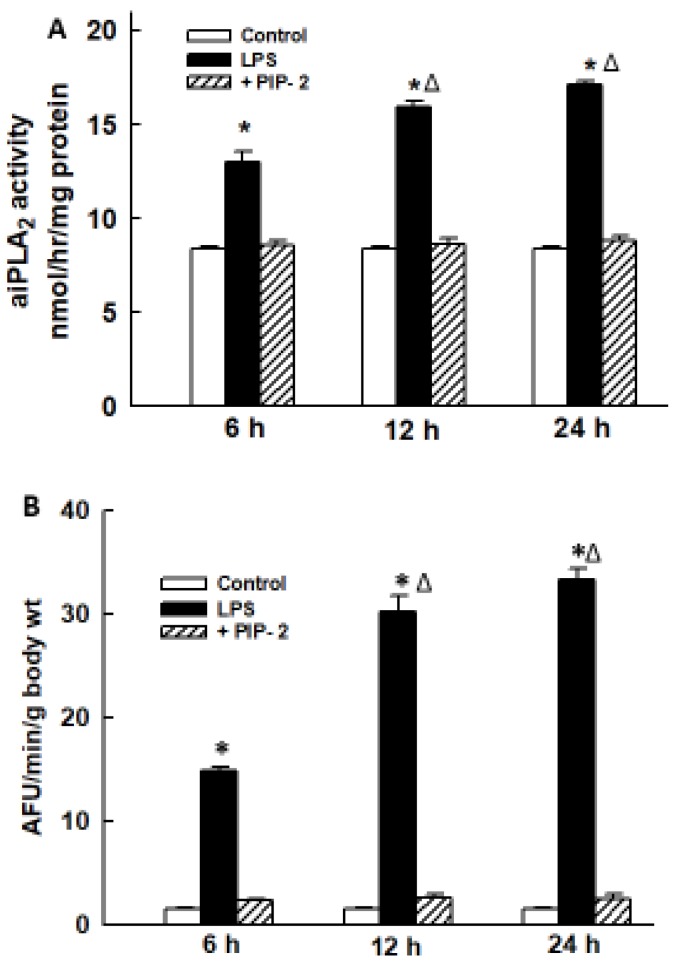
PIP-2 inhibits the increased lung aiPLA2 activity and increased ROS generation after LPS administration. LPS (5 µg/g body weight) was administered by intratracheal (IT) instillation along with liposomes alone (labeled as LPS) or with PIP-2 in liposomes (labeled as +PIP-2). Control was liposomes alone without LPS (labeled as control). Mice were sacrificed at 6, 12, or 24 h after LPS and lungs were perfused in situ for 15 min with saline solution containing the fluorophore difluorofluoroscein diacetate (DFF-DA). Lungs were then homogenized and assayed for (**A**) aiPLA_2_ activity; and (**B**) fluorescence of the lung homogenate as an index of ROS production. Results are mean ± SE for *n* = 3 for (**A**) and *n* = 4 for (**B**). * *p* < 0.05 vs. corresponding control and corresponding +PIP-2 values at the same time point; ^Δ^
*p* < 0.05 vs. the corresponding value at 6 h.

**Figure 3 ijms-20-02395-f003:**
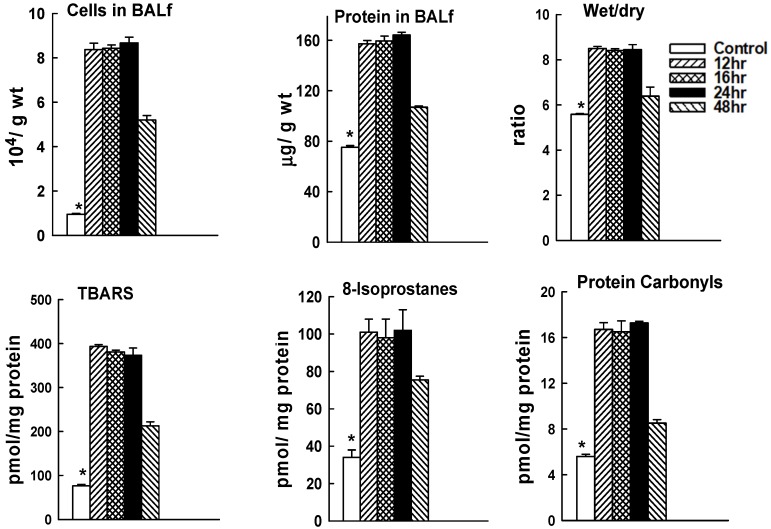
Mouse lung injury with LPS. C57Bl/6J were given IT LPS at 5 µg/g body weight. Mice were sacrificed at 12, 16, 24, or 48 h after LPS and lungs were evaluated for nucleated cells and protein in the bronchoalveolar lavage fluid (BALf), wet to dry weight ratio of the left upper lung lobe, and TBARS, 8-isoprostanes, and protein carbonyls in the lung homogenate. Results are mean ± SE for *n* = 4. * *p* < 0.05 vs all other values.

**Figure 4 ijms-20-02395-f004:**
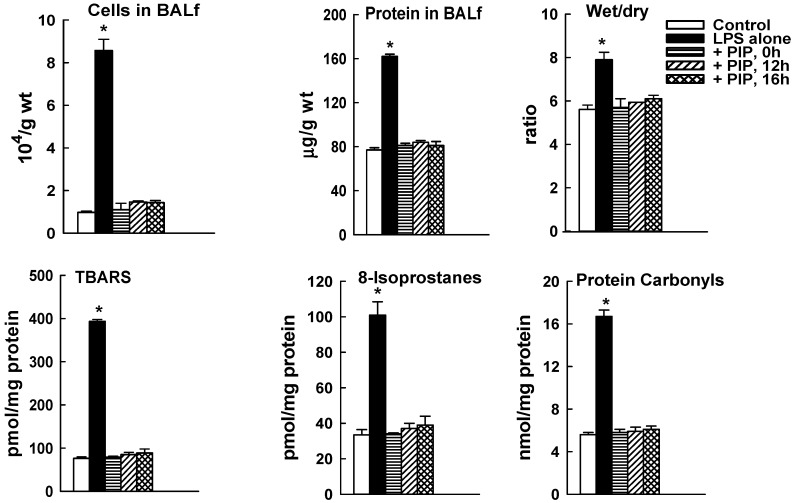
PIP-2 prevents mouse lung injury evaluated at 24h after IT LPS. LPS (5 µg/g body weight) was administered IT. PIP-2 was given IT coincidentally with LPS (0 time) or IV at 12 or 16 h after LPS. Mice were sacrificed at 24 h after LPS and lungs were evaluated for nucleated cells and protein in the bronchoalveolar lavage fluid (BALf), wet to dry weight ratio of the left upper lung lobe, and TBARS, 8-isoprostanes, and protein carbonyls in the lung homogenate. Results are mean ± SE for *n* = 4. * *p* < 0.05 vs. all other values.

**Figure 5 ijms-20-02395-f005:**
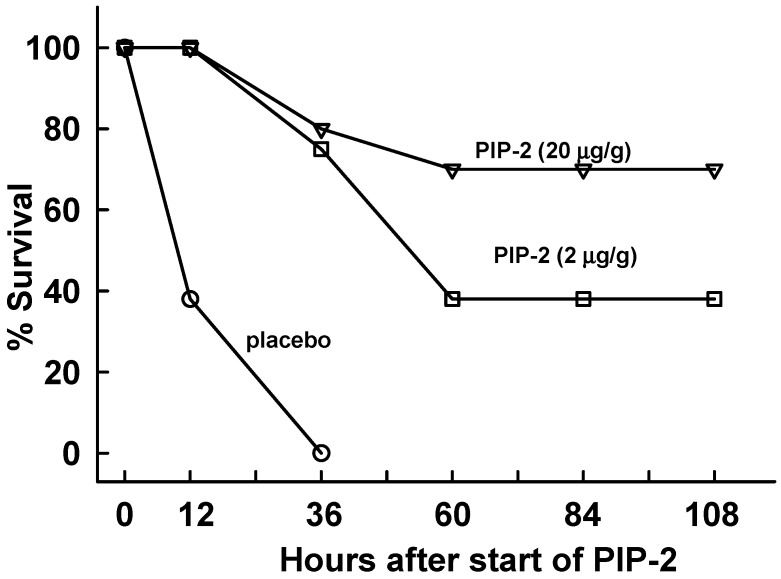
PIP-2 prevents mouse mortality with high dose LPS. Mice were administered LPS (15 µg/g body weight) by intratracheal instillation and divided into two groups. At 12 h after LPS, one group was given PIP-2 in liposomes by intravenous injection (IV) while the other group (placebo) was given liposomes alone. The time of the treatment initiation (12 h after LPS) is plotted as zero time. Treatment was repeated at 12, 36, 60, and 84 h after the initial dose of PIP-2,co-incident with the plotted points. Surviving mice were sacrificed at 108 h. *n* = 12 for placebo and *n* = 11 for PIP-2.

**Figure 6 ijms-20-02395-f006:**
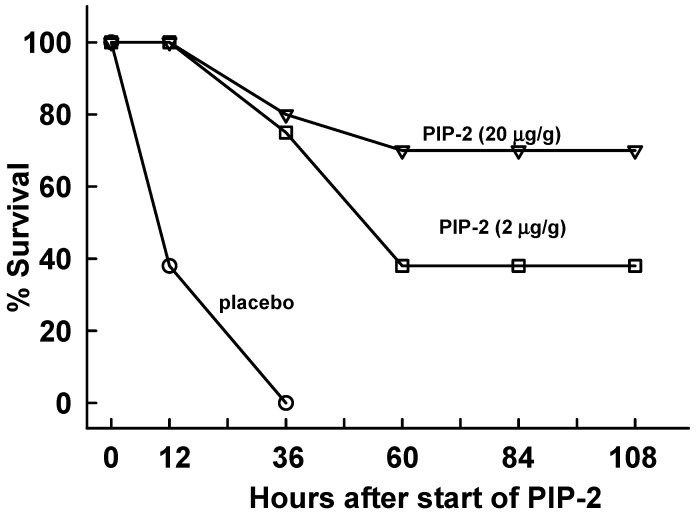
PIP-2 increases survival in mice given intraperitoneal LPS. Mice were administered LPS (15 µg/g body weight) by intraperitoneal (IP) injection and were 12 h later were divided into three groups. The placebo group (*n* = 8) was given IV liposomes only (no PIP-2). The two other groups were given PIP-2 in liposomes by IV injection, at either 2 µg/g body weight (*n* = 8) or 20 µg/g body weight (*n* = 10). Similar to the protocol described in Figure 5, the time of treatment initiation (at 12 h after LPS) was plotted as zero time.The dose of PIP-2 was repeated 12 h later and then 3 subsequent doses were gived at 24 h intervals (36,60,and 84 h after the start of PIP-2), co-incident with the plotted points. Surviving mice were sacrificed 24 h later at 108 h after start of treatment (120 h after administration of LPS).

**Table 1 ijms-20-02395-t001:** Lung injury is repaired in mice that survive high dose LPS.

	BALf Cells ×10^4^/g Body wt.	BALf Protein µg/g wt.	Wet/Dry Ratio	TBARS pmol/mg prot	8-Isoprostanes pmol/mg prot	Protein Carbonyls nmol/mg prot
Control (no LPS)	0.95 ± 0.04	75 ± 1.3	5.59 ± 0.03	76 ± 6	34 ± 3	5.6 ± 0.2
LPS + PIP-2	0.96 ± 0.40	78 ± 2.2	5.34 ± 0.03	77 ± 1	34 ± 3	5.6 ± 0.2

Mice were instilled IT with LPS (15 µg/g wt); PIP-2 (2 µg/g body wt) in liposomes was injected (IV) at the times indicated in Figure 5. Five of the surviving mice were sacrificed at 108 h after the start of treatment (120 h after LPS administration); control mice were given liposomes but not LPS. BALf, bronchoalveolar lavage fluid; TBARS, thiobarbituric reactive substances. Values are mean ± SE for *n* = 4 for control and *n* = 5 for LPS + PIP-2. None of the mean values for LPS + PIP-2 are statistically different (*p* > 0.05) from the corresponding control.

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
