# Peer review of "A Peptide Inhibitor of NADPH Oxidase (NOX2) Activation Markedly Decreases Mouse Lung Injury and Mortality Following Administration of Lipopolysaccharide (LPS)"

_ijms, 2019, doi:10.3390/ijms20102395_

Round 1

Reviewer 1 Report

LPS-induced lung injury and toxemia have been well studied. This manuscript reports the use of PIP-2 peptides to reduce the toxicity (mainly ROS production) and lung injury and thereby  to increase the survival rates of mice given LPS. 

The major concern is that  the quality of data presentation is poor. No marker of statistical significance in figures and no statistical methods are presented. For multiple groups, the student t test is not suitable. ANOVA test must be used. The Survival Curve is not the right form and no statistical analysis. The language expression and the format (particularly the space) need to be improved.

Figure legends are too short and did not give sufficient information to readers. The methods are also not in details, for example where the PIP-2 was purchased and how it prepared as well as how efficient the PIP-2 enter the cells etc. The half time in circulation and the why use the current dose and route as well as the time interval to give the PIP-2 to mice are not clearly justified.

Author Response

1)No marker of statistical significance in figures and no statistical methods are presented. 

   We now present the methods and indicate statistical significance in the Figures.

2)The Survival Curve is not the right form.

   We have replotted the data to the correct form (Kaplan-Meier).

3)Figure legends are too short and did not give sufficient information to readers.

   We have expanded the legends.

4)The methods are also not in details, 

 We have expanded the detail in the METHODS section.

5)how efficient the PIP-2 enter the cells

We showed previously that PIP-2 injected by itself does not inhibit PLA2 activity but encapsulation in liposomes does result in activity.Our conclusion was that PIP-2 by itself does not cross cell membranes,but liposomes gets it into cells.In the present mms,we have clarified the use of liposomes with PIP-2.

6)The half time in circulation

We found that the 1/2 time for PLA2 inhibition in the lung is ~50 hrs [ref Antioxidants (Basel) 7, 2018].. We have not measured 1/2 time in the circulation

7)why use the current dose and route

we have now explained our reasoning in the RESULTS section of the mms.

8)the time interval to give the PIP-2 to mice

we have also explained our choice of time interval

All of the comments were appropriate and useful and we thank you for pointing them out.

Reviewer 2 Report

This manuscript describes functional effects of a peptide inhibitor of NADPH oxidase activation, based on a 9 amino acid sequence in surfactant protein A that inhibits activity of peroxiredoxin, in mouse lung injury due to LPS administration.  Pretreatment with this peptide prevented LPS evoked production of ROS based on accumulation of fluorescent DFFDA in perfused lungs, as well as other indices of lung injury including BAL cell numbers, protein, wet/dry weights, or oxidative injury as indicated by TBARS, 8-isoprostanes or protein carbonyls.  Treatment with peptide at 12 or 16 hours after LPS also effectively diminished these endpoints in lungs harvested 24 hours after LPS. Mice pretreated with the peptide inhibitor followed by additional IV doses at 12, and then 24 hours serially enjoyed a survival benefit (increased from 0 to ~80%) after high dose LPS.  Finally mortality secondary to administration of intraperitoneal LPS (mimic of non-pneumonia related sepsis) was mitigated in a dose dependent manner by post treatment with IV peptide starting at 12 hours. The investigators speculate this peptide may be useful in clinical ALI.

Major concerns:

1.       Amplex red studies were performed in isolated perfused lungs treated with Ang II for activation of NOX2.  While figure 1 shows increased ROS production (based on amplex red oxidation) in Ang II treated lungs that can be blocked by PIP-2, it does not address the question of the capacity of PIP-2 to block an ROS increase with LPS.  The authors state that LPS treated lungs are “difficult to perfuse”.  Lower doses of LPS as opposed to Ang II could be used to verify increased ROS production in the model of interest if this is a significant problem.

2.       Figure 2 shows time dependent increases in DFFDA fluorescence in lung homogenates from LPS treated lungs, which is limited by pretreatment with PIP-2.  However, figure 3 shows an increase in wet to dry weights of the lungs and protein in the BAL, raising the probability that increased leakiness induced by LPS is responsible for the DFFDA fluorescence rather than increased ROS production.  At the very least, the impact of increased leakiness should be discussed.

3.       Although the functional importance of NADH oxidase inhibitors in LPS induced lung injury is established, this manuscript is largely descriptive in nature.  Several loops could be closed that are not, thereby providing more data to link treatment with PIP-2 to specific signaling pathways in lung cells.  For example, why not measure lung aiPLA2 with LPS with PIP-2 or liposomal controls?  Why not measure Rac activation under the same conditions?

4.       Student’s t test are not appropriate to compare repeated measures over time. There are no indicators of significance in the figures.

Minor concerns:

1.       The introduction is unnecessarily long and not all directly related to the studies being presented. It could be shortened.  

2.       The discussion describes reversal of PMN infiltration, but technically, only BAL was examined.  The authors should either grade lung histology or change the working of their discussion.

3.       The text states that physical activity, food and water consumption and body habitus were improved when mice receiving LPS were treated with PIP-2.  Each of these endpoints can be quantitated. The data should be provided if possible.

The last sentence of the discussion should probably be eliminated.  The authors correctly note the potential danger in inhibited ROS production in the sentences above.  It does not necessarily follow that the relatively short period for treatment of a brief disease (weeks for recovery or death) would render alterations in physiological function evoked by peptides unimportant. 

Author Response

1) The Amplex red study shows that PIP-2 inhibits NOX2 mediated ROS production. Fig 2 shows that LPS activates ROS and that is also inhibited by PIP-2. We have deleted the irrelevant statement about perfusion of LPS treated lungs.

2)We know of no mechanism that an increase in lung water would increase lung DFF fluorescence. Lung edema might quench (probably slightly) the fluorescence and we have now mentioned that possibility.

3) Yes,we think that measurements of aiPLA2 activity would add to the mms and have added them as Fig 2A. Thanks for the suggestion.We are not in a position to measure Rac and we have already published our study [FASEB J 30: 2885-98, 2016] showing that Rac provides  the connection between increased aiPLA2 activity and NOX2 activation.

4) Yes,we screwed up on the statistics part.We definitely used ANOVA to test significance and have corrected the Methods.We have also now indicated significance in Figures and Table.

Minor

1) We have shortened the Intro slighty, but are reluctant to cut too much since many readers may not be 'lung injury' literate.

2) Yes, we agree that PMN may leave the BAL but still be present in the lung as the reviewer indicated.We have reworded the Discussion to take that into account.

3) We are not in a position to repeat these expts to quantitate food and water consumption--so we have deleted that observation.We attach several videos to show return of physical activity in PIP-2 treated mice..

4) We have deleted the last sentence as suggested.

Round 2

Reviewer 1 Report

Much better.

Should have a document for the answers to reviewers' comments

Reviewer 2 Report

The manuscript is improved with the changes and is now acceptable for publication.